# Code-Switching with Word Senses for Pretraining in Neural Machine Translation

**Vivek Iyer**[1]    **Edoardo Barba**[2]    **Alexandra Birch**[1]    **Jeff Z. Pan**[1]    **Roberto Navigli**[2]

[1]School of Informatics, University of Edinburgh
[2]Sapienza NLP Group, Sapienza University of Rome
{vivek.iyer, pinzhen.chen, a.birch}@ed.ac.uk
{barba, navigli}@diag.uniroma1.it

## Abstract

Lexical ambiguity is a significant and pervasive challenge in Neural Machine Translation (NMT), with many state-of-the-art (SOTA) NMT systems struggling to handle polysemous words (Campolungo et al., 2022a). The same holds for the NMT pretraining paradigm of denoising synthetic "code-switched" text (Pan et al., 2021; Iyer et al., 2023), where word senses are ignored in the noising stage – leading to harmful sense biases in the pretraining data that are subsequently inherited by the resulting models. In this work, we introduce Word Sense Pretraining for Neural Machine Translation (WSP-NMT) - an end-to-end approach for pretraining multilingual NMT models leveraging word sense-specific information from Knowledge Bases. Our experiments show significant improvements in overall translation quality. Then, we show the robustness of our approach to scale to various challenging data and resource-scarce scenarios and, finally, report fine-grained accuracy improvements on the DiBiMT disambiguation benchmark. Our studies yield interesting and novel insights into the merits and challenges of integrating word sense information and structured knowledge in multilingual pretraining for NMT.

## 1 Introduction

Lexical ambiguity is a long-standing challenge in Machine Translation (Weaver, 1952) due to polysemy being one of the most commonly occurring phenomena in natural language. Indeed, thanks to a plethora of context-dependent ambiguities (e.g. the word *run* could mean *run a marathon*, *run a mill*, *run for elections* etc.), words can convey very distant meanings, which may be translated with entirely different words in the target language. To deal with this challenge, traditional Statistical Machine Translation approaches tried to incorporate Word Sense Disambiguation (WSD) systems in MT with mostly positive results (Carpuat and Wu, 2007; Xiong and Zhang, 2014). These were followed by similar efforts to plug sense information in NMT frameworks (Liu et al., 2018; Pu et al., 2018). But, since the introduction of the Transformer (Vaswani et al., 2017), the task of disambiguation has largely been left to the attention mechanism (Tang et al., 2018, 2019).

In the last three years, though, many works have challenged the ability of modern-day NMT systems to accurately translate highly polysemous and/or rare word senses (Emelin et al., 2020; Raganato et al., 2020; Campolungo et al., 2022a). For example, Campolungo et al. (2022a) expose major word sense biases in not just bilingual NMT models like OPUS (Tiedemann and Thottingal, 2020), but also commercial systems like DeepL[1] and Google Translate[2], and massively pretrained multilingual models (Tang et al., 2021; Fan et al., 2021). A likely explanation is that these models capture inherent data biases during pretraining. This particularly holds for the pretraining paradigm of denoising code-switched text[3] — most notably, Aligned Augmentation (AA, Pan et al., 2021), where, during the pretraining phase, input sentences are noised by substituting words with their translations from multilingual lexicons, and NMT models are then tasked to reconstruct (or 'denoise') these sentences. AA and subsequent works (Reid and Artetxe, 2022; Iyer et al., 2023; Jones et al., 2023) show the benefits of code-switched pretraining for high- and low-resource, supervised and unsupervised translation tasks. Despite their success, a major limitation of these substitution mechanisms is that they are unable to handle lexical ambiguity adequately, given their usage of *'sense-agnostic'* translation lexicons. In fact, in most of these works, substitutions for polysemes are chosen randomly, regardless of context (Pan et al., 2021;

---

[1]https://www.deepl.com/en/translator
[2]https://translate.google.com
[3]In this work, we refer to this family of approaches as 'code-switched pretraining' for brevity

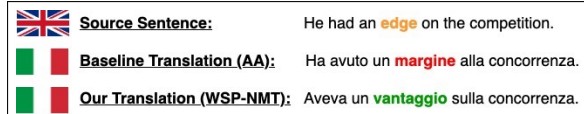

Figure 1: Italian translations of a sentence from the DiBiMT disambiguation benchmark (Campolungo et al., 2022a) by: a) our main baseline, Aligned Augmentation (AA, Pan et al., 2021), and b) our approach, WSP-NMT. AA mistranslates the ambiguous word edge as margine *(border, rim)*. Due to 'sense-pivoted pretraining', WSP-NMT correctly translates it as vantaggio *(advantage)*.

Reid and Artetxe, 2022; Jones et al., 2023).

In an effort to introduce knowledge grounding at the word sense level during pretraining and potentially minimise data errors, enhance convergence, and improve performance, we propose the notion of **'sense-pivoted pretraining'** – to move code-switched pretraining from the *word level* to the *sense level*. Specifically, we propose an approach called Word Sense Pretraining for Neural Machine Translation (WSP-NMT) that first disambiguates word senses in the input sentence, and then code-switches with sense translations for denoising-based pretraining. Figure 1 provides an intuition of how integrating disambiguation in pretraining helps our model handle ambiguous words better, avoiding defaulting to more frequent senses, and reducing errors in translation.

Indeed, our experiments on using WSP-NMT yield significant gains in multilingual NMT – about +1.2 spBLEU and +0.02 COMET22 points over comparable AA baselines in high-resourced setups. Among other interesting performance trends, we observe that our margin of improvement increases substantially as we move towards low-resource (+3 to +4 spBLEU) and medium-resource (+5 spBLEU) settings. Lastly, for more fine-grained evaluation, we also compare our models on the DiBiMT disambiguation benchmark (Campolungo et al., 2022a) for Italian and Spanish, and note accuracy improvements of up to 15% in the challenging task of verb disambiguation.

Our key novel contributions are, thus, as follows:

1. We show how incorporating WSD in NMT pretraining can outperform the widely used paradigm of lexicon-based code-switching.

2. We demonstrate how reliable structured knowledge can be incorporated into the multilingual pretraining of NMT models, leading to error reduction and improved performance.

3. We evaluate the robustness of WSP-NMT to scale to various challenging data and resource-constrained scenarios in NMT, and point out its efficacy in low-resource and zero-shot translation tasks.

4. Finally, we evaluate the disambiguation capabilities of our models on the DiBiMT benchmark, and contribute a fine-grained understanding of the scenarios where WSP-NMT helps resolve lexical ambiguity in translation.

## 2 Related Work

**Multilingual pretraining for NMT** The success of multilingual NMT in the latest WMT shared tasks (Akhbardeh et al., 2021; Kocmi et al., 2022) has heightened the research focus on noising functions used in the denoising-based multilingual pretraining of NMT models. While masking has been successful in scaling to massive models (Tang et al., 2021; Costa-jussà et al., 2022), there has also been a parallel strand of research exploring more optimal noising functions that grant superior performance at lower data quantities. In particular, code-switched pretraining (Yang et al., 2020; Lin et al., 2020; Pan et al., 2021) has gained popularity since it moves the denoising task from language modelling to machine translation, and induces superior cross-lingual transfer. Notably, Pan et al. (2021) proposed Aligned Augmentation that code-switches text using MUSE lexicons, and trained the mRASP2 model – yielding SOTA results on a wide variety of translation tasks, beating strong baselines pretrained on more data. Subsequent research has tried to extend this: Li et al. (2022) and Reid and Artetxe (2022) combine AA-like lexicon-based code-switching with masking, while Iyer et al. (2023) and Jones et al. (2023) explore optimal code-switching strategies. While effective, none of these works have attempted to move code-switching to the word sense level. Most of them sample random sense translations when encountering polysemes, with these errors further propagating during pretraining. Li et al. (2022) attempt to partially circumvent this issue by choosing the appropriate sense translation based on the reference sentence, but this trick cannot scale to code-switching (abundant) monolingual data or for code-switching in languages other than the target one. Iyer et al. (2023) use pretrained translation and word alignment models to provide empirical gains, but the reliability is low with these black-

box systems since such techniques lack grounding. In order to handle lexical ambiguity in a principled manner, we propose WSP-NMT to provide knowledge grounding at the word sense level while pretraining multilingual NMT models.

**WSD for Machine Translation** In the traditional Statistical Machine Translation paradigm, incorporating WSD was shown to improve translation quality (Carpuat and Wu, 2007; Chan et al., 2007; Xiong and Zhang, 2014). With the rise of NMT, various techniques were proposed to integrate word sense information. Nguyen et al. (2018) simply provided annotations of Korean word senses to an NMT model, while other works then computed sense embeddings using tricks like Neural Bag-of-Words (Choi et al., 2017), bidirectional LSTMs (Liu et al., 2018) and adaptive clustering (Pu et al., 2018) – using these to augment word embeddings in the training of sequence-to-sequence models (Bahdanau et al., 2015). With the introduction of the Transformer (Vaswani et al., 2017), Tang et al. (2019) hypothesized that Transformer encoders are quite strong at disambiguation, and used higher-layer encoder representations to report sense classification accuracies of up to 97% on their test sets.

However, with the creation of challenging disambiguation benchmarks more recently (Raganato et al., 2019, 2020; Campolungo et al., 2022a), the purported capabilities of NMT models have been called into question once again. Indeed, when top-performing open-source and commercial systems were evaluated on the DiBiMT benchmark (Campolungo et al., 2022a), it was found that even the best models (i.e. DeepL and Google Translate) yielded accuracies < 50% when translating ambiguous words, biasing heavily towards more frequent senses, and vastly underperformed compared to the (then) SOTA WSD system, ESCHER (Barba et al., 2021a) – indicating significant room for improvement. While recent works have tried to address this by fine-tuning models with sense information (Campolungo et al., 2022b), we explore if we can avoid an extra fine-tuning stage and incorporate WSD information while pretraining itself, so as to yield enhanced off-the-shelf performance.

**Structured Knowledge for Machine Translation** Though limited, there have been some scattered efforts to use Knowledge Graphs (KGs) in MT. In bilingual NMT, Zhao et al. (2020b) generate synthetic parallel sentences by inducing relations

between a pair of KGs, while Moussallem et al. (2019) augment Named Entities with KG Embeddings, comparing RNNs and Transformers. Lu et al. (2019) and Ahmadnia et al. (2020) enforce monolingual and bilingual constraints with KG relations and observe small gains for bilingual English-Chinese and English-Spanish pairs respectively. It remains unclear how these constraint-based methods would scale to the current paradigm of pretraining multilingual NMT models. Relatedly, Hu et al. (2022) proposed multilingual pretraining leveraging Wikidata, but their sole focus was on Named Entity (NE) translation. They showed that their approach performs comparably overall, while improving NE translation accuracy. Our work is complementary to theirs and shows how a KG could be used to pretrain a multilingual NMT model with generic concept translations, improving overall NMT as well as disambiguation performance.

## 3 Approach

### 3.1 Definitions

**Code-switching** is defined as the phenomenon of shifting between two or more languages in a sentence. In this work, it refers to the 'noising' function used for synthesising pretraining data.

**Word Sense Disambiguation** is the task of linking a word $w$ in a context $C$ to its most suitable sense $s$ in a predefined inventory $I$ and can thus be represented as a mapping function $f(w, C, I) = s$. In this work, we use BabelNet (Navigli et al., 2021) as our reference sense inventory.

### 3.2 Aligned Augmentation

Aligned Augmentation (AA, Pan et al., 2021) is a denoising-based pretraining technique that uses MUSE lexicons[4] $M$ for code-switching the source sentence. These lexicons provide non-contextual, one-to-one word translations. If a word has multiple possible translations (e.g. polysemes), AA (and subsequent works) randomly choose one regardless of context – making this approach 'sense-agnostic'. Moreover, in order to code-switch in multiple languages, Pan et al. (2021) also chain bilingual lexicons together, by pivoting through English, and this causes a further propagation of 'sense-agnostic' errors. Nevertheless, AA achieved SOTA results in massively multilingual NMT, so we use it as our primary baseline and aim to verify

---

[4] https://github.com/facebookresearch/MUSE

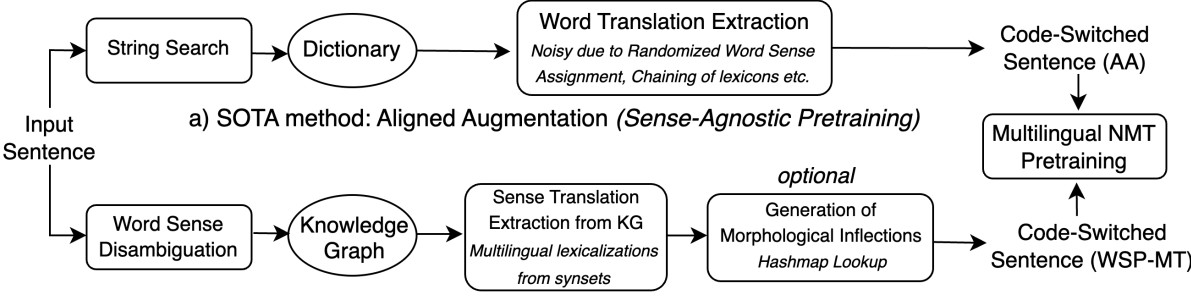

Figure 2: Pipelines contrasting the approach used by the AA and WSP-NMT pretraining algorithms.

the impact of 'sense-pivoted' pretraining by comparing against similarly trained WSP-NMT models. We also note that while some extensions to AA have been proposed recently, by combining it with masking or other code-switching strategies (see Section 2), these improvements likely also hold for code-switching with sense translations – making these techniques complementary to our work.

## 3.3 Word Sense Pretraining for NMT

We depict the WSP-NMT pipeline, and compare it with AA, in Figure 2. It involves three stages:

### 3.3.1 Word Sense Disambiguation (WSD)

To determine the most appropriate translation for a word $w_i$ in a sentence, we begin by identifying its intended meaning in context using WSD systems. To measure the impact of disambiguation quality on the final translation performance, we adopt two WSD systems, namely, ESCHER, a SOTA open-inventory WSD model, and AMuSE-WSD (Orlando et al., 2021), a faster but still competitive, off-the-shelf classification system. We note that, since ESCHER was originally only released for English, we also trained models for other languages on the multilingual datasets in XL-WSD (Pasini et al., 2021). We provide further training details in Appendix A.2. As a result of this stage, each content word in the input sentence is linked to a specific synset[5] in BabelNet.

### 3.3.2 Extracting Translations from BabelNet

Given the synset associated with a disambiguated word, we retrieve its possible translations in the languages of experimentation by using BabelNet. Indeed, in BabelNet, each synset in each language comes with a set of lexicalizations that can express the concept represented by that synset. At the moment of writing, BabelNet contains concept lexical-

izations for 520 languages with varying degrees of coverage. Thus, at the end of this stage, all content words will have a list of possible translations in the target languages.

### 3.3.3 Generation of Morphological Inflections

Finally, given that concept lexicalizations in BabelNet are present as lemmas, we include an additional postprocessing stage to convert target lemmas $l_i$ of sense $s$ into their contextually appropriate morphological variant $m_i$. We do this by leveraging MUSE lexicons $M$. We first preprocess entries in $M$ using a lemmatizer $L$ like so: given word translation pairs $(x_j, y_j)$ in $M$, we create a hashmap $H$ where $H(x_j, L(y_j)) = y_j$. Given $H$, during code-switching, if the source word $w_i$ is not a lemma we do a lookup for the key $H(w_i, l_i)$ and return the corresponding value as the contextually appropriate morphological inflection $m_i$ for sense $s$ – which is then used for code-switching. If the key is unavailable in $H$, we code-switch $w_i$ with $l_i$ itself.

This step allows us to generate translations that take into account both word senses and morphology – thus combining the best of both worlds between 'sense-agnostic' lexicons and canonicalized[6] Knowledge Graphs storing sense information. While this postprocessing does improve performance, it is optional and we show in Section 4.2 that baselines without this stage also yield decent improvements, while minimizing the overhead. We use Trankit (Nguyen et al., 2021) for lemmatization, which was shown to achieve SOTA results on this task on treebanks from 90 languages[7].

### 3.4 Training

For fair comparison, we follow the experimental setup proposed by Pan et al. (2021) for training our

---

[5]A synset is a group of semantically equivalent word senses.

[6]i.e. KGs representing concepts as canonicalized lemmas.
[7]https://trankit.readthedocs.io/en/latest/performance.html

| Monolingual | | | | | | Parallel | | | |
|---|---|---|---|---|---|---|---|---|---|
| **En** | 7.5M | Fr | 7.5M | **It** | 7.5M | **En-Es** 1.8M | **En-It** | 1.7M | |
| **Es** | 7.5M | Pt | 7.5M | **Ro** | 7.5M | **En-Fr** 1.8M | **En-Ro** | 365K | |

(a) Size of Romance language corpora (sentence count)

| Monolingual | | | | | | Parallel | |
|---|---|---|---|---|---|---|---|
| **En** | 20M | **Hi** | 6M | **Fa** | 6M | **En-Hi** 1.9M | **En-Fa** 1M |

(b) Size of Indo-Iranian language corpora (sentence count)

Table 1: Monolingual and Parallel Data statistics.

AA and WSP-NMT models. We code-switch parallel and monolingual sentences (with translations generated using the respective approaches) and shuffle them to create our training dataset $D$. We input the code-switched sentences to the encoder, while the target sentences are the reference sentences for parallel data and the original 'denoised' sentences for monolingual data. We prepend a special language token indicating language ID to each input and target sentence. Finally, we train the model using a loss function $\mathcal{L}$ that jointly optimizes Contrastive Loss $\mathcal{L}_{\text{CON}}$ and Cross Entropy Loss $\mathcal{L}_{CE}$. In principle, Contrastive Loss minimizes the semantic representation gap between positive examples (code-switched and reference sentences) and maximizes the distance between negative examples (approximated to other random samples in the minibatch). With this in mind, we define $\mathcal{L}$ as:

$$\mathcal{L} = \mathcal{L}_{CE} + |s| * \mathcal{L}_{CON}$$

$$\text{where: } \mathcal{L}_{CE} = - \sum_{(x,y) \in D} \log P_\theta(y|x), \text{and}$$

$$\mathcal{L}_{\text{CON}} = - \sum_{(x,y) \in D} \log \frac{e^{\text{sim}^+(\mathcal{E}(x), \mathcal{E}(y))/\tau}}{\sum_{(a,b) \in B} e^{\text{sim}^-(\mathcal{E}(x), \mathcal{E}(b))/\tau}}$$

Here, $\mathcal{E}$ signifies the average pooled encoder representations for the input sentence, while 'sim' computes positive and negative semantic similarities between the contrastive examples, denoted by $\text{sim}^+$ and $\text{sim}^-$ respectively. Temperature $\tau$ controls the difficulty of distinguishing between positive and negative samples, and is set to 0.1. $B$ denotes the mini-batch in dataset $D$ that $(x, y)$ belongs to. Lastly, $|s|$ is the average token count of sentence $s$ that balances token-level cross entropy loss and sentence-level contrastive loss.

| Language Pair | En-Es | En-Fr | En-It | En-Ro |
|---|---|---|---|---|
| **Test Set** | WMT'13 | WMT'14 | WMT'09 | WMT'16 |
| **Language Pair** | **En-Pt** | | **En-Hi** | **En-Fa** |
| **Test Set** | FLORES 200 | | WMT'19 | FLORES 200 |

Table 2: Test sets used per language in this work.

| Model | En | It | Fr | Es | Avg | Running Time (for 330K sents) |
|---|---|---|---|---|---|---|
| AMuSE-WSD | 77.13 | 76.03 | 80.35 | 72.77 | 76.57 | **78 mins** |
| ESCHER | **78.72** | **79.14** | **83.94** | **77.52** | **79.83** | 180 mins |

Table 3: F1-scores of ESCHER and AMuSE-WSD on XL-WSD (Pasini et al., 2021) for English (En), Italian (It), French (Fr), Spanish (Es), and the Average (Avg) of all scores. All results are statistically significant ($p < 0.01$ as per McNemar's test (Dieterich, 1998)). For comparing efficiencies, we also report the running time for disambiguating 330K sentences on 1 RTX 3090.

## 4 Experiments

### 4.1 Experimental Setup

Table 1 shows the statistics for our training data. In this work, we primarily experiment with the Romance languages and pretrain multilingual NMT models on the monolingual and parallel data in Table 1a, following the training setup described previously. We use Portuguese later for our zero-shot experiments, so no parallel data is provided. We also explore NMT for Indo-Iranian languages in Section 4.3.3, and use the data given in Table 1b to train those models. We note here that since our objective in this work is to explore the utility of KGs for sense-pivoted pretraining, our data setups follow a similar scale as other works on KG-enhanced NMT (Zhao et al., 2020a; Xie et al., 2022; Hu et al., 2022), and diverges from the massively multilingual paradigm of Pan et al. (2021).

For validation, we sample the last 1000 sentences from the parallel corpus per pair. As shown in Table 2, for testing, we use either the latest WMT test sets per pair, if available, or FLORES 200 sets (Costa-jussà et al., 2022) otherwise. We evaluate NMT results using spBLEU[8] (Costa-jussà et al., 2022), chrF++[9] (Popović, 2017) and COMET22 (Rei et al., 2022) metrics, and compute statistical significance using Student's T-test (Student, 1908).

We use the Transformer (Vaswani et al., 2017) architecture with 6 encoder and 6 decoder layers

---

[8] `nrefs:1|case:mixed|eff:no|tok:flores101|smooth: exp|version:2.3.1`

[9] `nrefs:1|case:mixed|eff:yes|nc:6|nw:2|space: no|version:2.3.1`

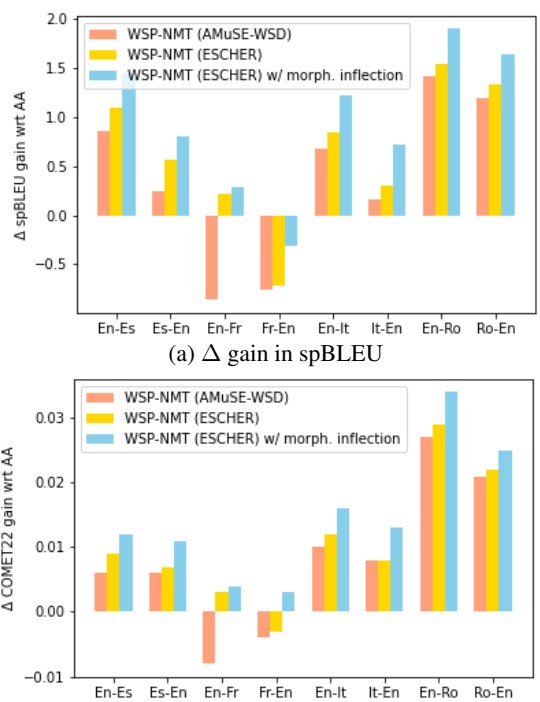

(a) Δ gain in spBLEU

(b) Δ gain in COMET22

Figure 3: Improvements in the average performance of WSP over AA for each language pair.

for training our models. For tokenization, we use sentencepiece (Kudo and Richardson, 2018). We train the sentencepiece model on concatenated and shuffled data from Table 1, using a unigram language model with a vocabulary size of 32,000 and character coverage of 1.0. We use replacement ratios of 0.1 and 0.05 for our Romance and Indo-Iranian models respectively. We provide further hyperparameter settings in Appendix A.1.

Lastly, in Table 3, we evaluate and report key performance statistics for our two WSD systems, AMuSE-WSD and ESCHER. We shall use these to better interpret our NMT results and comment on the observed quality/speed tradeoff in Section 4.2.

## 4.2 Main Results

We show overall results for the Romance language experiments, averaged across 5 runs, in Table 4. We also report language pair-specific improvements in Figure 3. We make the following key observations:

1. **WSP-NMT consistently outperforms AA.** We observe that our best WSP-NMT model, i.e. the one using ESCHER for disambiguation and MUSE lexicons for generating morphological predictions, consistently beats AA models by significant margins across all metrics – including a +1.2 boost in spBLEU for En-X and +0.7 for X-En. Except for Fr-En where WSP-NMT

1. And when Federer levelled the match, one-time British No 1 Petchey **temía** *(feared->fear->temer->temía)* the worst.

2. The world was **mirando** *(looking->look->mirar->mirando)* at him.

3. GM **deber** *(owes->owe->debe->deber)* $23.4bn, excluding $3.9bn in realised Treasury **pérdidas** *(losses->loss->pérdida->pérdidas)*, according to a Treasury statement dated Friday.

Figure 4: WSP-NMT with Morphological Inflection. Code-switched words are highlighted. `a'->a->b->b'` denotes the inflection generation process explained in Section 3.3.3 where `a'` and `b'` are source and target words, while `a` and `b` are the corresponding lemmas. `a->b` is extracted from BabelNet, while the hashmap lookup yields `b'` given (`a'`, `b`).

marginally underperforms[10], these gains also extend to individual language pairs (Figure 3). This suggests that minimising errors in pretraining data with 'sense-pivoted pretraining' can enhance overall convergence and yield stronger models.

2. **Superior disambiguation helps, but cheaper WSD systems work well too.** We observe in Figure 3 that ESCHER consistently induces greater gains than AMuSE-WSD across all pairs. This makes sense given the superior disambiguation quality of the former (Table 3). However, we note that AMuSE-WSD, which is about 2.3x faster, also outperforms AA and can be used as an effective, but cheaper alternative, suitable for disambiguating large corpora under budget constraints.

3. **Postprocessing KG lemmas to predict the relevant morphological inflection is beneficial.** We observe generating morphological inflections by intersecting with MUSE lexicons (Section 3.3.3) yields major boosts across the board. We understand this qualitatively by showing examples of such inflections from our dataset, in Figure 4. We observe that this approach helps in ensuring linguistic agreement (such as for tense and number) which, in turn, enhances the quality of code-switching and, thus, pretraining. Our technique can thus be used as an effective way to bridge the gap between KG lemmas and morphologically inflected words in corpora – a gap which has proved to be one of the major roadblocks in leveraging structured knowledge in NMT so far.

4. **Higher gains are observed for the lower resourced En-Ro pair.** WSP-NMT obtains a +2

---

[10]This is possibly because the WordNet that BabelNet uses for French is WOLF (Sagot and Fišer, 2008), an automatically constructed silver quality resource

| Baseline | En-X | | | X-En | | |
|---|---|---|---|---|---|---|
| | spBLEU | chrF++ | COMET22 | spBLEU | chrF++ | COMET22 |
| AA | 24.385 ± 0.639 | 47.890 ± 0.647 | 0.714 ± 0.009 | 24.725 ± 0.577 | 49.660 ± 0.453 | 0.716 ± 0.006 |
| WSP-NMT (AMuSE-WSD) | 24.910 ± 0.336 | **48.525 ± 0.304** | **0.723 ± 0.006** | 24.935 ± 0.313 | 49.880 ± 0.243 | **0.724 ± 0.005** |
| WSP-NMT (ESCHER) | **25.310 ± 0.909** | 48.765 ± 0.717 | **0.728 ± 0.013** | 25.095 ± 0.903 | 49.905 ± 0.705 | 0.725 ± 0.010 |
| WSP-NMT (ESCHER) + morph. inflection | **25.595 ± 0.279** | **49.115 ± 0.235** | **0.731 ± 0.003** | **25.435 ± 0.179** | **50.130 ± 0.229** | **0.729 ± 0.002** |

Table 4: Overall results of the Romance language experiments with respect to spBLEU, chrF++ and COMET22 metrics. Mean and standard deviation after 5 runs, initialised with different random seeds (0-4) has been reported. Statistically significant improvements ($p \leq 0.05$) over our primary baseline, AA, are highlighted in bold.

spBLEU improvement over AA for the En-Ro pair – which is the lowest resourced in our setup, with 365K parallel sentences (Table 1). This suggests our approach could be even more useful for medium and low data setups – something we empirically verify in Section 4.3.1.

5. **More significant boosts are obtained for COMET22 than string-based spBLEU or chrF++ metrics.** While our best model improves across all the metrics, we observe this in Table 4 for our weaker models. This is probably because the errors made by incorrect word sense translations can sometimes be too subtle to be adequately penalized by weaker string-based metrics. On the other hand, neural metrics like COMET22 are more robust and have been trained to detect word-level as well as sentence-level errors. It has shown high correlations with human evaluation, ranking second in the WMT'22 Metrics shared task (Freitag et al., 2022). We, thus, conclude that statistically significant gains in COMET scores, when averaged across 5 seeds, help in increasing the strength and reliability of our results.

### 4.3 How well does WSP-NMT extend to more challenging MT scenarios?

Here, we try to evaluate the robustness of our approach to scale to various challenging translation scenarios: a) data scarcity, b) zero-shot translation and c) translation of under-represented languages.

#### 4.3.1 Scaling to data-scarce settings

In Figure 5, we explore how performance varies with the scale of parallel data, while keeping monolingual data constant – since the latter is usually more abundant. We observe an improvement margin of +3 to +4 spBLEU in the low-resource setting (50-100K sentences) that widens to +5 spBLEU in the medium-resource setup (250-500K sentences). The gap then narrows as we move towards high-

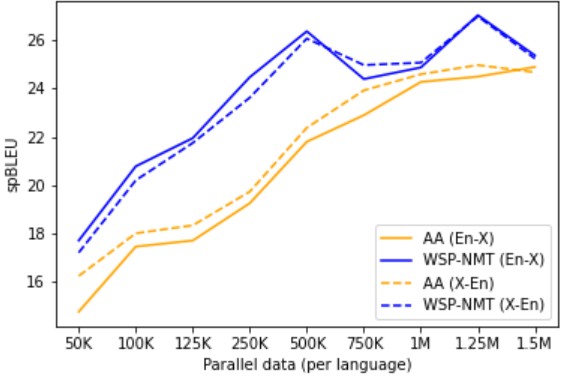

Figure 5: Average En-X and X-En spBLEU scores for AA and WSP-NMT models, with variation in parallel data (in terms of maximum sentence count per language)

| Baseline | En-Pt | Pt-En |
|---|---|---|
| AA | 2.92 ± 0.64 | 6.88 ± 1.56 |
| WSP-NMT (best) | **3.60 ± 0.19** | **8.52 ± 0.53** |

Table 5: Zero-Shot Translation scores (spBLEU) for the English-Portuguese (En-Pt) pair. We use the best WSP-NMT model from Table 4 and compare with AA. Statistical significance (p=0.04) is highlighted in bold.

resource scenarios (750K-1.5M sentences), suggesting AA models become more robust to noise when high quantities of parallel data are provided. Based on these findings, we discuss potential applications of our method in Section 4.5.

#### 4.3.2 Scaling to Zero-Shot Translation

Continuing our exploration into data scarcity, we now evaluate the performance of WSP-NMT with no parallel data (i.e. zero-shot) in Table 5. We use En-Pt as a simulated zero-shot pair, following the setup in Table 1a. We compare our best WSP-NMT model from Table 4 with AA, averaged across 5 runs as before. We observe statistically significant gains of about +0.7 and +1.64 spBLEU for En-Pt and Pt-En pairs respectively. We conclude that WSP-NMT is strong in a wide range of scenarios

|  | En-X | X-En |
|---|---|---|
| AA | 22.79 ± 1.063 | 20.49 ± 0.893 |
| WSP-MT (AMuSE-WSD) | 22 ± 0.704 | 19.58 ± 0.559 |
| WSP-MT (AMuSE-WSD) + morph. inflection | 21.83 ± 0.727 | 19.53 ± 0.72 |
| WSP-MT (AMuSE-WSD) + synonyms + hypernyms | 22.71 ± 0.93 | 20.23 ± 0.906 |

Table 6: spBLEU scores for Indo-Iranian languages (Hindi and Farsi) to English, averaged across 5 seeds

and can improve performance at varying degrees of resourcedness, including in zero-shot tasks.

### 4.3.3 Scaling to under-represented languages

Lastly, we explore how WSP-NMT performs for languages under-represented in external resources such as WSD systems, lexicons, and KGs. The issues faced here include low coverage, poor quality, and unreliable WSD. To study the impact of these factors (in isolation from data scarcity), we train a multilingual NMT model between English and 2 Indo-Iranian languages (Hindi and Farsi) using the data in Table 1b. We choose these languages since they are under-represented in BabelNet[11], are supported by AMuSE-WSD[12] and have MUSE lexicons available for training AA baselines.

In Table 6, we find that integrating morphological inflection does not yield gains, likely because – due to poor coverage – we find matches in MUSE lexicons far fewer times (only 18.3% compared to 74% for the Romance data). BabelNet also suffers from poor coverage (45% of disambiguated senses do not have translations in either Hindi or Farsi). To address this, we explore if translations of synonyms and hypernyms of a KG concept could be used as substitutes. In practice, we find that, while this does improve the vanilla WSP-NMT model, it is unable to beat AA. This is likely rooted in the fact that low-resource disambiguation is an unaddressed challenge in NLP, primarily because of a lack of resources – be it training datasets, reliable WSD models, or even evaluation benchmarks. Despite these hurdles, we note that, since WSP-NMT is effective in data-scarce settings, creating a small amount of sense-annotated data in these languages might suffice to yield potential gains.

---

[11] https://babelnet.org/statistics
[12] https://nlp.uniroma1.it/amuse-wsd/api-documentation. We are unable to use ESCHER here due to lack of relevant training data in XL-WSD.

### 4.4 DiBiMT benchmark results

Finally, we test our models on the DiBiMT disambiguation benchmark (Campolungo et al., 2022a) in Figure 6. The DiBiMT test set comprises sentences with an ambiguous word, paired with "good" and "bad" translations of this word. Then, given an NMT model's hypotheses, it computes accuracy, i.e. the ratio of sentences with "good" translations compared to the sum of "good" and "bad" ones. Hypotheses that fall in neither of these categories are discarded as 'MISS' sentences. We compare AA and our best WSP-NMT model on Italian and Spanish, the two Romance languages supported by DiBiMT, and report overall accuracies. In addition, we also include fine-grained scores on disambiguation of specific POS tags like nouns and verbs to better understand the capabilities of our models.

We observe a small increase in overall accuracy. Looking more closely, we find that while WSP-NMT's accuracy for nouns is comparable to AA, significant improvements are observed for verb disambiguation – especially for Italian, where average accuracy increases from 15.34% to 17.65% (15% increase). Prior research has shown that verbs are much harder to disambiguate than nouns since they are highly polysemous (Barba et al., 2021a,b; Campolungo et al., 2022a) and require much larger contexts to disambiguate (Wang et al., 2021). Thus, while sense-agnostic AA models are able to disambiguate nouns, verb disambiguation proves more challenging, which is where sense-pivoted pretraining with WSP-NMT helps. We provide example translations in Appendix A.3 to drive intuition of this further. Finally, we note that WSP-NMT also reduces the average MISS% - from 39.3% to 37.2% for Spanish, and 41.2% to 40.2% for Italian. These trends are encouraging, since errors such as word omission, word copying and hallucination are the most common issues and constitute about 50% of all 'MISS' cases (Campolungo et al., 2022a).

### 4.5 Applications

We conclude our results section by noting that while WSP-NMT is not ideally suited for either very low-resource (Section 4.3.3) or very resource-rich (Section 4.3.1) settings, it does yield significant performance gains for low and medium-data settings of well-resourced (e.g. European) languages, as shown in Figure 5. We hypothesize that this can, thus, be quite useful for domain-specific translation in these languages, such as in news,

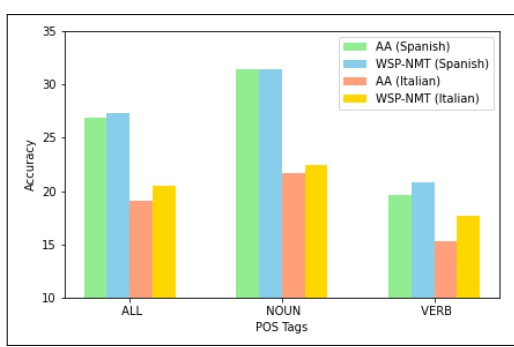

Figure 6: Accuracy of our AA and WSP-NMT models, in Spanish and Italian, on the DiBiMT benchmark.

healthcare, e-commerce, etc. In such information-centric domains, minimising disambiguation errors and improving translation quality with reliable, general-purpose KGs would likely bring value.

## 5 Conclusion

We propose WSP-NMT, a 'sense-pivoted' algorithm leveraging word sense information from KBs for pretraining multilingual NMT models. We observe significant performance gains over the baseline, AA, and note various factors of interest including the impact of WSD quality and morphology. We then study how WSP-NMT scales to various data and resource-constrained scenarios and, finally, note significant accuracy gains in verb disambiguation on the DiBiMT benchmark. Our results help emphasize the utility and strength of WSP-NMT.

## Limitations

While our approach attempts to minimize the risk of low-quality translations creeping into the pretraining data by using human-curated structured knowledge, it is possible certain biased translations enter our data nonetheless. For instance, during code-switching, there is a possibility of gender bias in the sense translations chosen, as certain gendered translations of a concept that is ungendered in the source language are provided in the same word sense in BabelNet. Though we use random sampling to mitigate such risks, the propagation of potential biases depends heavily on the fairness and diversity of translations available in BabelNet. While such issues are similar to those that would arise in the current paradigm of dictionary-based code-switching, a principled approach to resolve gender based on sentence-level context and to choose the appropriate translation from the BabelNet synset

would be a useful direction of future research. The BabelNet KG also has the potential for improving in this regard, allowing for the addition of gender categories to specific concepts.

We also note that WSP-NMT may not be as effective in the translation of under-represented languages. As we note in Section 4.3.3, this is partly rooted in the unavailability of disambiguation resources and high-quality Knowledge Bases for such languages. However this is quickly changing: i) WSD for low-resource languages has been a very emerging area of research recently (Saeed et al., 2021; Bhatia et al., 2022; Boruah, 2022), and, at the same time, ii) BabelNet itself is constantly being updated and scaled up. For example, due to the creation of newer low-resource wordnets, BabelNet expanded from 500 supported languages (v5.1 from Jul 2022) to 520 (v5.2 from Nov 2022) in just 4 months. As the quality of these curated resources improves with community contributions and advances in low-resource NLP/WSD, we anticipate that methods such as WSP-NMT, which we show to be robust in low-data settings (<1M parallel sents), could be increasingly useful.

## Acknowledgements

This work was funded by UK Research and Innovation (UKRI) under the UK government's Horizon Europe funding guarantee [grant number 10039436]. The authors also gratefully acknowledge the support of the PNRR MUR project PE0000013-FAIR.

The computations described in this research were performed using the Baskerville Tier 2 HPC service (https://www.baskerville.ac.uk/). Baskerville was funded by the EPSRC and UKRI through the World Class Labs scheme (EP/T022221/1) and the Digital Research Infrastructure programme (EP/W032244/1) and is operated by Advanced Research Computing at the University of Birmingham.

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

| Hyperparameter | Romance | Indo-Iranian |
|---|---|---|
| Batch Size | 4000 | 4000 |
| Learning Rate | 0.0001 | 0.0001 |
| Update Frequency | 4 | 4 |
| Optimizer | Adam w/ eps=5e-07 | Adam w/ eps=1e-08 |
| Weight Decay | 0.0001 | 0.01 |
| Dropout | 0.05 | 0.01 |
| Label Smoothing | 0.1 | 0.1 |
| Contrastive Lambda | 1.0 | 1.0 |
| Temperature | 0.1 | 0.01 |
| Clip Norm | 10 | 10 |
| Replacement Ratio | 0.1 | 0.05 |

Table 7: Hyperparameters used for training our models.

| Model | En | It | Fr | Es | Params | Running Time (for 100K sents) |
|---|---|---|---|---|---|---|
| ESCHER | 78.72 | 79.14 | 83.94 | 77.52 | 692 M | 63 mins |
| ConSeC | 79.00 | 79.30 | 84.40 | 77.40 | 287 M | 213 mins |

Table 8: Results of four different ESCHER models trained on English (En), Italian (It), French (Fr) and Spanish (Es) and ConSeC.

# A Appendix

## A.1 Hyperparameters

We provide our detailed hyperparameter settings in Table 7. We optimize each hyperparameter independently in separate runs, and use the best settings in our experiments.

## A.2 WSD Systems Details

In the original AMuSE-WSD work, a multilingual version of the system was made available; however, ESCHER currently offers only an English model. For this reason, we train three distinct ESCHER models specifically tailored for Italian, Spanish, and French. We adopt the methodology outlined in the original work but adapt it to train the models using the XL-WSD (Pasini et al., 2021) splits for the respective languages and use mDeBERTa (He et al., 2023) as the underlying multilingual transformer architecture. By doing so, we ensure compatibility and improve performance in the targeted linguistic contexts as reported in Barba et al. (2021b) without incurring prohibitive computational costs. As a reference, we report in Table 8 the difference in performance between our ESCHER models and the results reported in Barba et al. (2021b) for the multilingual version of their system, ConSeC.

## A.3 Verb Disambiguation Examples

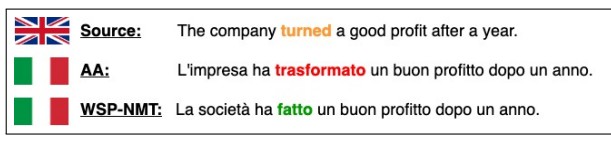

(a) "trasformato" means "transformed" and is an incorrect sense translation of "turned". "fatto" means "made", as in "made a good profit" and is a correct translation in context.

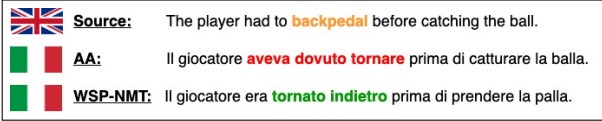

(b) "adeguare" is a verb meaning "adapt" or "adjust" and is an incorrect sense translation of "appropriate". "stanziare" means "allocate", as in "allocate funds", and is correct in context.

(c) "aveva dovuto tornare" translates to "had to return" and is an incorrect translation of "backpedal". "tornato indietro" means to "move (or run) back" and contextually correct.

Figure 7: Examples of verb disambiguation in DiBiMT Italian hypothesis translations by AA and WSP-NMT.

We show some examples of improved verb disambiguation of WSP-NMT in Figure 7, using examples from the DiBiMT test set. The AA hypotheses for these sentences contain translations that could have been appropriate in certain contexts but are incorrect here. For example, in Figure 7a, turned could translate into transformato when the context is referring to "turning into something". A similar explanation holds for Figure 7b where appropriate could translate into adeguare when talking about "appropriating style", "appropriating language" or "appropriating software", to name a few examples. Figure 7c is very interesting since backpedal does mean tornato indietro ("move back") in this context, but "tornare" ("return") is not an appropriate sense translation. These cases reflect how by carefully modelling for cross-lingual convergence of word senses during pretraining, WSP-NMT can avoid subtle translation errors that may arise when encountering ambiguous words.