# OpenReview forum: "Code-Switching with Word Senses for Pretraining in Neural Machine Translation"
_EMNLP/2023/Conference — EMNLP 2023 Findings_

### Official Review · Reviewer_MJt2 · 2023-08-05

**Soundness:** 4

**Excitement:**

3: Ambivalent: It has merits (e.g., it reports state-of-the-art results, the idea is nice), but there are key weaknesses (e.g., it describes incremental work), and it can significantly benefit from another round of revision. However, I won't object to accepting it if my co-reviewers champion it.

**Paper Topic And Main Contributions:**

This paper analyzes the shortcomings of existing methods in solving the polysemy problem, with the main issue being that "they are unable to handle lexical ambiguity adequately, given their usage of ‘sense-agnostic’ translation lexicons." In order to address this issue, the authors propose a sense-pivoted pretraining method, which shifts codes-switched pretraining from the word level to the sense level, leveraging word sense-specific information from Knowledge Bases. Experimental results on WMT'13 and FLORES 200, by training a transformer with 12 layers, demonstrate that the proposed method can enhance translation quality, especially in medium and low resource settings. The improvement in high-resource settings is 1.2 spBLUE and 0.02 COMET22, in medium-resource settings it's 5 spBLUE, and in low-resource settings it's 3-4 spBLUE.

Contribution：
1. Proposal of a novel code-switched pretraining method, integrating Word Sense Disambiguation into NMT pretraining.
2. Comprehensive experiments are conducted to evaluate the performance of the proposed method.
3. The authors also evaluate their method on the DiBiMT benchmark, which measures the disambiguation capabilities.

**Questions For The Authors:**

1. Can you provide more potential explanations as to why your method might underperform compared to the baseline?
2. When using larger or smaller models, do you believe the conclusions proposed in this paper remain consistent?

**Reasons To Accept:**

1. The quality of writing is high and reader-friendly. The paper is well-organized.
2. The idea presented is logical and easy to understand.
3. Comprehensive experiments are conducted to verify the effectiveness of the proposed method in different scenarios, such as different language pairs across high, medium, and low resources.

**Reasons To Reject:**

1. For some experimental results, there is a lack of reasonable and sufficient explanation. For instance, in figure 3, the authors' method underperforms the baseline in the en-fr and fr-en settings. The reason and analysis for this are missing.
2. A majority of the experiments focus on the presentation of results. The analyses of the method itself and the experimental outcomes are not comprehensive enough. Given that the authors' method underperforms the baseline in some instances, one might question to what extent the performance improvement brought by this pretraining method can be attributed to the authors' claim of "moving code-switched pretraining from the word level to the sense level, by leveraging word sense-specific information from Knowledge Bases".

**Reproducibility:**

4: Could mostly reproduce the results, but there may be some variation because of sample variance or minor variations in their interpretation of the protocol or method.

**Reviewer Confidence:**

3: Pretty sure, but there's a chance I missed something. Although I have a good feel for this area in general, I did not carefully check the paper's details, e.g., the math, experimental design, or novelty.

---

> ### Author Rebuttal · Authors · 2023-08-28
>
> Thank you for this glowing review! We are pleased that you too found the writing quality of our work high, our idea logical and easy to understand and our experiments comprehensive. We address your concerns below:
>
> ## Reasons to Reject
>
> 1. Unlike the statistically significant improvements in other language pairs, the performance drop for En-Fr and Fr-En is mostly statistically insignificant (only the AMuSE WSD baseline is significantly worse). As for why significant improvements were not observed for French, this is possibly because of the relatively poorer quality of BabelNet in this language. The WordNet that BabelNet uses for French is WOLF [1], which is an automatically constructed silver-quality resource.
>
> We promise to clarify both these points in Section 4.2, and also to indicate the statistically significant changes in Figure 3.
>
> 2. Actually, we do analyze the contribution of the various subcomponents of our method in Table 4 by verifying the role of WSD quality (with AMuSE and ESCHER) and that of morphological inflection.
>
> Moreover as promised to R1, in order to provide deeper analysis of our results, we will be conducting disambiguation probing of encoder representations of models pretrained using AA and WSP-NMT respectively using a linear WSD classifier. This will help clarify whether our method encodes sense-specific features when compared to sense-agnostic pretraining. We will include these results in the extra page of the camera-ready version.
>
> ## Questions to the Author
>
> 1. Answered just above, thanks for pointing this out!
>
> 2. We had not tried this before but, on your suggestion, we just tried this during the rebuttal phase and observed similar improvements. We trained Transformer-large models for AA and WSP-NMT with Escher WSD (ie. 12 encoders and 12 decoders, instead of 6 and 6 as used in our work), using the same data setup and the same random seed (seed=0). We show our average scores below:
>
> AA: 27.55 spBLEU (En-X) and 28.25 spBLEU (X-En)
>
> WSP-NMT:  28.80 spBLEU (En-X) and 28.90 spBLEU (X-En)
>
> This translates to an average improvement of +1.25 spBLEU and +0.65 spBLEU for En-X and X-En respectively. En<->Ro still gains the most -- with +2.2 spBLEU and +1.7spBLEU for En-Ro and Ro-En respectively. We shall add these findings in the camera-ready version, along with the results of the WSP-NMT baseline with morphological inflection (which we could not try now due to time constraints, but would likely yield even higher gains).
>
> ## References
>
> [1] Sagot and Fišer, 2008. Building a free French wordnet from multilingual resources

---

### Official Review · Reviewer_sGfj · 2023-08-07

**Soundness:** 4

**Excitement:**

4: Strong: This paper deepens the understanding of some phenomenon or lowers the barriers to an existing research direction.

**Paper Topic And Main Contributions:**

What problem or question this paper addresses
+ There is a method that learns denoise of sentences that were swapped with other language words using a bilingual dictionary. When different meaning words are used to swap because of polyseme, it leads to harmful sense biases.

Method
+ The authors estimate synsets using WSD. Then they obtain words in other languages using a knowledge graph. The obtained words in other languages are used to swap words in the training data for a denoising task.


The main contributions (Strengths)
+ This method is a straight and reasonable solution to the problem.
+ The experimental results show the effectiveness of the method.

Weaknesses
+ This method uses a WSD method, synsets, and a knowledge graph. Therefore, this method cannot be applied to the languages or domains for which these resources are not available.


**Reasons To Accept:**

See the main contributions.

**Reasons To Reject:**

See the weaknesses in Paper Topic And Main Contributions.

**Reproducibility:**

4: Could mostly reproduce the results, but there may be some variation because of sample variance or minor variations in their interpretation of the protocol or method.

**Reviewer Confidence:**

4: Quite sure. I tried to check the important points carefully. It's unlikely, though conceivable, that I missed something that should affect my ratings.

---

> ### Author Rebuttal · Authors · 2023-08-28
>
> Thank you so much for your high scores and appreciation of our work! We are pleased you found our approach easy and intuitive and our experimental results effective.
>
> Regarding your concern of our technique’s applicability for low-resource languages, we do agree this is challenging at the moment. But, as we point out to Reviewer 4nWC, low-resource disambiguation is an active and emerging field of research currently [1][2][3][4]. Simultaneously, BabelNet itself is constantly expanding and its quality is constantly being refined. As the quality of both disambiguators and curated resources improves with the research community contributions and advances in low-resource NLP, we anticipate that methods such as ours that are shown to be robust in low-data settings (<1M parallel sents) would be increasingly useful.
>
> ### References
>
> [1] Rouhizadeh et al., 2020, Knowledge-based word sense disambiguation with distributional semantic expansion for the Persian language
>
> [2] Saeed et al., 2021, Investigating the Feasibility of Deep Learning Methods for Urdu Word Sense Disambiguation
>
> [3] Bhatia et al., 2022, Role of Genetic Algorithm in Optimization of Hindi Word Sense Disambiguation
>
> [4] Boruah, 2022, A Novel Approach to Word Sense Disambiguation for a Low-Resource Morphologically Rich Language

---

### Official Review · Reviewer_4nWC · 2023-08-11

**Soundness:** 4

**Excitement:**

3: Ambivalent: It has merits (e.g., it reports state-of-the-art results, the idea is nice), but there are key weaknesses (e.g., it describes incremental work), and it can significantly benefit from another round of revision. However, I won't object to accepting it if my co-reviewers champion it.

**Missing References:**

N/A

**Paper Topic And Main Contributions:**

This paper proposes a novel pre-training strategy for multilingual NMT models called "Word Sense Pretraining for Neural Machine Translation (WSP-NMT)". Recent pre-training paradigms have centered on Aligned Augmentation (AA) [1], which allows pre-training by denoising a few tokens in the source sentences substituted by sense-agnostic multilingual lexicons, thus limiting to fully solve lexical ambiguity. Therefore, the authors shift the code-switched paradigm from sense-agnostic pre-training towards sense-pivoted pre-training.

The WSP-NMT pipeline consists of three stages: 1) Word sense disambiguation (WSD), 2) Extracting sense translation, 3) Transforming morphological infections. For each stage, they leverage the appropriate SOTA models and knowledge base.

Finally, they effectively demonstrate their improvements in lexical ambiguity resolution on the DiBiMT benchmark dataset along with the BLEU measure.

[1] Pan et al., 2021 Contrastive Learning for Many-to-many Multilingual Neural Machine Translation


**Questions For The Authors:**

1. To validate whether WSP-NMT learns a better representation space, there could be a comparison experiment that sense-appropriate sentence pairs are similar than sense-mismatched sentence pairs.

2. Continuing with Question 1, there could be room for applying a contrastive loss between sense-appropriate sentence pairs and sense-mismatched sentence pairs.
(It's acknowledged that learning the difference between positive and negative samples will be difficult.)


**Reasons To Accept:**

1. The motivation of this paper is clear.
- The authors target the core weakness of the existing code-switched pre-training strategy, which struggles to address lexical ambiguity by generating noise with sense-agnostic lexicons.

2. The authors show their achievements in resolving lexical ambiguity effectively.
- It's especially noteworthy for the performance gains on verb inflections, which are difficult to disambiguate due to the plethora of polysemous words.

**Reasons To Reject:**

1. The experimental configuration in terms of high, medium, and low resource settings slightly departs from other studies.
- The authors set 100\~200K, 500K\~1M, 1.5M\~3M as low, medium, and high resource settings based on parallel corpus size. However, in the PC32 dataset paper [2] used by the previous work [1], which the authors follow the experimental setting, much larger datasets were used as rich-source. Considering Figure 3 and 5 (if x-axis is scaled to a huge size of dataset), it might be unable to beat AA [1] in high-resource setting.

2. The applicability of WSP-NMT is questionable.
- The authors claim that WSP-NMT outperforms on low-resource language pairs. Since many data-scarce language pairs are under-represented in WSD models and BabelNet, there might be several situations where it is difficult to utilize WSP-NMT.

[2] Lin et al., 2020, Pre-training Multilingual Neural Machine Translation by Leveraging Alignment Information


**Reproducibility:**

3: Could reproduce the results with some difficulty. The settings of parameters are underspecified or subjectively determined; the training/evaluation data are not widely available.

**Reviewer Confidence:**

3: Pretty sure, but there's a chance I missed something. Although I have a good feel for this area in general, I did not carefully check the paper's details, e.g., the math, experimental design, or novelty.

**Typos Grammar Style And Presentation Improvements:**

It would be recommended to revise the expression "posit", which is widely used by Chat-GPT but awkward in Point 5 of 4.2 Main Results.

---

> ### Author Rebuttal · Authors · 2023-08-28
>
> Thank you so much for your detailed and comprehensive review. We are pleased you found our work well-motivated and our experiments sound and noteworthy. The concerns you raise are highly interesting and we feel are worth discussing in some depth, so we have put forth our view below:
>
>
> ## Reasons to Reject
>
>
> We do agree that it can be challenging to apply WSP-NMT to rich-resource or very low-resource languages.
>
> 1. We see your point, but actually, our data setup follows a similar scale as recent works on KG-enhanced NMT [1][2][3][4] that use data quantities ranging anywhere between 200K-2M parallel sentences. We diverge from the data setup of Pan et al., 2021, since unlike them we do not seek to train a massively multilingual NMT model. We shall clarify these points in the paper.
>
> Regarding rich-resource languages, while we are limited by computational constraints, it is possible that training on huge amounts of parallel data diminishes the gains offered by our approach. Such a result would align with previous works on KG-enhanced NMT – where it has been observed that while knowledge grounding does help with increased reliability, quality, and task-specific performance in quite a few cases, overall results (such as BLEU scores) improve with massive scale regardless.
>
>
> 2. Even on the applicability of WSP-NMT for low-resource languages, we agree with you: i) reliable disambiguators are not yet available and ii) knowledge bases for those languages are still not comparable to rich-/medium-resource ones. However this is quickly changing: i) WSD for low-resource languages has been a very emerging area of research recently [5][6][7][8], and, at the same time, ii) BabelNet itself is constantly being updated and scaled up. For example, due to the creation of newer low-resource WordNets, BabelNet expanded from 500 supported languages (v5.1 from Jul 2022) to 520 (v5.2 from Nov 2022) in just 3 months. As the quality of these curated resources improves with community contributions and advances in low-resource NLP/WSD, we anticipate that methods such as WSP-NMT, which we show to be robust in low-data settings (<1M parallel sents), could be increasingly useful.
>
>
> We conclude by noting that while WSP-NMT is not ideally suited for either of these extreme settings, it does appear to be beneficial for low and medium-data settings of well-resourced (eg. European) languages. This can be quite useful for domain-specific translation in these languages, such as in news, healthcare, e-commerce etc. In such information-centric domains, minimising sense-agnostic errors and improving translation quality with reliable, general-purpose KGs would likely be of value.
>
> This is what we try to convey in L494-L499, but we will make this clearer in the camera-ready version and shall add a short paragraph of discussion on potential applications of our method.
>
> ## Questions to the Author:
>
> Very interesting suggestions! Incidentally, both of these questions are being planned as part of future work.
>
> 1. We agree that analyzing the representation space induced by WSP-NMT will make our paper more interesting. We will add the results of a disambiguation probing experiment in the extra page of the camera-ready version, wherein we shall probe encoder representations of ambiguous words, using models trained by AA and WSP-NMT algorithms. To conduct the probing, we shall train a simple linear classifier on these representations using the standard framework for WSD evaluation [9] and compare test accuracy to verify whether our pretraining strategy improves the sense-awareness of the final representation space.
>
>
> 2. We were originally planning to try this version of contrastive loss as well. However, we realized a potential issue with considering only other senses of the word as negative examples is that it would minimize entropy and probably affect how general sentence-level representations are learnt. A middle ground would be to take a good mixture of sentences with opposite senses as well as random sentences as negative examples. Another option is to take a weighted sum of these two contrastive losses. We are considering these directions, along with a form of curriculum learning (for learning general representations first and sense-specific ones later), as part of future work. While these should further improve our results, we felt this was outside the scope of our current work, which sought to give a proof of concept of sense-pivoted pretraining.
>
> ## Grammar improvements
>
> Thank you - as a principle we have not used chatGPT in this work. Sorry about this wording though, we shall correct this.
>
>
> ## References
>
> [1] Zhao et al., 2020  Knowledge Graph Enhanced Neural Machine Translation via Multi-task Learning on Sub-entity Granularity
>
> [2] Zhao et al., 2021 Knowledge Graphs Enhanced Neural Machine Translation
>
> [3] Xie et al., 2022 End-to-end entity-aware neural machine translation
>
> [4] Hu et al., 2022 DEEP: DEnoising Entity Pre-training for Neural Machine Translation
>
> [5] Rouhizadeh et al., 2020, Knowledge-based word sense disambiguation with distributional semantic expansion for the Persian language
>
> [6] Saeed et al., 2021, Investigating the Feasibility of Deep Learning Methods for Urdu Word Sense Disambiguation
>
> [7] Bhatia et al., 2022, Role of Genetic Algorithm in Optimization of Hindi Word Sense Disambiguation
>
> [8] Boruah, 2022, A Novel Approach to Word Sense Disambiguation for a Low-Resource Morphologically Rich Language
>
> [9] Raganato et al., 2022, Word sense disambiguation: a unified evaluation framework and empirical comparison

---

### Meta-Review · Area_Chair_h3by · 2023-09-18

**Recommendation:** 4

**Metareview:**

This paper proposes a new method that consists in using Word Sense Disambiguation along with Knowledge Graphs during pretraining to address the translation of polysemous words.
The problem is highly relevant for the community and the method precisely address weaknesses of current methods.
The paper is clear and well written, the experimentations are well conducted and the results demonstrate the effectiveness of the proposed method, even if some additional analysis would sometimes be beneficial. The authors mentioned in their rebuttal that most of the missing analysis will be added in the camera ready version.
The main concern is the usability of the approach for very low and very high resource languages, but this is not a criteria to reject the paper per se.

---

### Decision · Program_Chairs · 2023-10-07

**Decision:**

Accept-Findings

**Comment:**

This paper proposes a new method that consists in using Word Sense Disambiguation along with Knowledge Graphs during pretraining to address the translation of polysemous words.
The problem is highly relevant for the community and the method precisely address weaknesses of current methods.
The paper is clear and well written, the experimentations are well conducted and the results demonstrate the effectiveness of the proposed method, even if some additional analysis would sometimes be beneficial. The authors mentioned in their rebuttal that most of the missing analysis will be added in the camera ready version.
The main concern is the usability of the approach for very low and very high resource languages, but this is not a criteria to reject the paper per se.